# miRNA- and Cell Line-Specific Constraints on Precursor miRNA Processing of Stably Transfected Pancreatic Cancer and Other Mammalian Cells

**DOI:** 10.3390/ijms25115666

**Published:** 2024-05-23

**Authors:** Taylor J. Allen-Coyle, Berta Capella Roca, Alan Costello, Niall Barron, Joanne Keenan, Martin Clynes, Fiona O’Neill, Finbarr O’Sullivan

**Affiliations:** 1The SFI Research Centre for Pharmaceuticals (SSPC), Bernal Institute, University of Limerick, V94 T9PX Limerick, Ireland; taylor.jade.allen96@gmail.com (T.J.A.-C.); fiona.oneill@dcu.ie (F.O.); 2SSPC Research Group, National Institute for Cellular Biotechnology, Life Sciences Institute, Dublin City University, D09 E432 Dublin, Ireland; 3Cell Engineering Group, National Institute for Bioprocessing Research and Training (NIBRT), A94 X099 Dublin, Ireland

**Keywords:** vector design, miRNA processing, precursor miRNA, stable transfections, miRNAs, pancreatic cancer

## Abstract

MicroRNAs (miRNAs) regulate approximately one-third of all human genes. The dysregulation of miRNAs has been implicated in the development of numerous human diseases, including cancers. In our investigation focusing on altering specific miRNA expression in human pancreatic cancer cells, we encountered an interesting finding. While two expression vector designs effectively enhanced miR-708 levels, they were unable to elevate mature forms of miR-29b, -1290, -2467, and -6831 in pancreatic cancer cell lines. This finding was also observed in a panel of other non-pancreatic cancer cell lines, suggesting that miRNA processing efficiency was cell line specific. Using a step-by-step approach in each step of miRNA processing, we ruled out alternative strand selection by the RISC complex and transcriptional interference at the primary miRNA (pri-miRNA) level. DROSHA processing and pri-miRNA export from the nucleus also appeared to be occurring normally. We observed precursor (pre-miRNA) accumulation only in cell lines where mature miRNA expression was not achieved, suggesting that the block was occurring at the pre-miRNA stage. To further confirm this, synthetic pre-miRNA mimics that bypass DICER processing were processed into mature miRNAs in all cases. This study has demonstrated the distinct behaviours of different miRNAs with the same vector in the same cell line, the same miRNA between the two vector designs, and with the same miRNA across different cell lines. We identified a stable vector pre-miRNA processing block. Our findings on the structural and sequence differences between successful and non-successful vector designs could help to inform future chimeric miRNA design strategies and act as a guide to other researchers on the intricate processing dynamics that can impact vector efficiency. Our research confirms the potential of miRNA mimics to surmount some of these complexities.

## 1. Introduction 

MicroRNAs (miRNAs) are a class of non-coding RNAs approximately 22 nucleotides (nt) in length that regulate gene expression [1]. Since their discovery in the nematode *Caenorhabditis elegans* in 1993 by the Ambros and Ruvkun group [2], members of this endogenous class of RNAs have received considerable attention due to their roles in diverse biological functions, including the regulation of cell development, differentiation, and survival [3]. Highlighting their importance, around one-third of all human genes are known to be regulated by miRNAs, with approximately 2469 miRNAs identified to date [4]. The dysregulation of miRNAs has been implicated in the development and progression of numerous human diseases, including cancers [5]. To investigate the function of miRNAs, we need robust methods for their overexpression, both in vitro and in vivo, that mimic natural mechanisms. Overexpression of miRNAs is achieved via one of two strategies: synthetic mature miRNAs or miRNA-expressing DNA vectors. DNA vectors can either be constitutively active or have inducible expression systems. Constitutively active plasmids may interfere with cellular functions, including cell death. Therefore, user-defined heterologous miRNA regulation is favoured. Inducible systems reveal the direct consequence of certain genetic manipulations and are especially beneficial when working with mammalian cell lines that are controlled by highly intricate genetic networks [6]. 

The canonical miRNA biogenesis pathway constitutes three steps: (1) Transcription and processing of primary miRNAs (pri-miRNAs) into hairpin-structured precursor miRNAs (pre-miRNAs) (~60–70 nucleotides) by the microprocessor complex, consisting of DROSHA (a ribonuclease III enzyme) and its cofactor, DGCR8 [3,7,8]; (2) Export of pre-miRNAs to the cytoplasm via the exportin 5 (XPO5)/RanGTP complex, followed by processing by DICER, an RNase III endonuclease. DICER recognizes the ends of pre-miRNA and cleaves the dsRNA stem, yielding a ~22 nucleotide mature miRNA duplex with 2 nt 3’ overhangs; (3) Loading of the guide strand of the mature miRNA duplex into Argonaute (AGO) proteins, forming an miRNA-induced silencing complex (miRISC). This complex, along with GW182 proteins, guides the miRISC to target mRNAs for silencing [3,8]. 

Dysfunctional microprocessing machinery has been previously described in cancer, where mutations due to XPO5 cause pre-miRNAs to accumulate in the nucleus, leading to decreased mature miRNA levels [2,9]. In addition, proteins frequently dysfunctional in cancer cells, such as p53, which also functions as a modulator in miRNA processing, could attenuate the efficient production of mature miRNAs [10]. Meanwhile, the processing machinery can become saturated via prolonged overexpression of miRNAs via expression vectors [11]. These findings suggest that impaired processing in cancer cells needs to be taken into great consideration when designing miRNA expression vectors. 

In previous studies from our group, we carried out a global comparison of miRNAs in human pancreatic cancer tumours, adjacent normal tissue, and matched patient-derived xenograft models (PDX) using microarray screening [12,13]. From this screen, we compiled a list of priority miRNAs that were overexpressed in both comparisons between normal and tumour and tumour and PDX. To investigate the role of these priority miRNAs, we generated stable overexpression vectors for each. This study addresses the obstacles in designing stable miRNA expression vectors for cancer cells. Despite using a similar construct as miR-708, overexpression vectors for miR-29b-1-5p, -1290, -2467-3p, and -6831-5p failed to elevate the levels of mature miRNAs above background endogenous levels in MIA PaCa-2 and PANC-1 cells. Due to the high levels of expression achieved by miR-708, it was chosen as a positive control for downstream experiments. In order to understand why these miRNAs were not expressed, we followed each of them through the sequential processing steps in MIA PaCa-2 cells while also investigating expression and processing in a number of other cell types.

## 2. Results 

### 2.1. RNA Secondary Structure and Primary Sequence Features Are Key Elements Responsible for Efficient miRNA Processing

From our initial list of priority miRNAs that can be found in our previous publications [12,13], initial attempts to overexpress miR-29b, -1290, -2467-3p, and -6831-5p using overexpression vectors (described in methods) were unsuccessful, leading to no detectable expression of mature miRNAs by RT-qPCR in either MIA PaCa-2 or PANC-1 cells as shown in the figure in Section 2.5. Compared to the four miRNAs listed above, miR-708-5p showed the highest levels of mature miRNA levels in MIA PaCa-2 (RQ = 4331) and PANC-1 (RQ = 419) and was chosen as the positive control miRNA for follow-up studies. Figure 1 displays the predicted secondary structure of the designed miRNAs with 70 base pair genomic flanking regions (referred to as design 1). As miR-708 was successfully expressed, and to eliminate the possibility that the secondary structures and/or primary sequence motifs of the four problem miRNAs were unsuitable for recognition by the microprocessing machinery, these four miRNAs were re-cloned into the genomic flanking region of miR-708 (referred to as design 2). The assessment of predicted secondary structures of designed miRNAs and their impact on miRNA processing involves multiple criteria. The predicted secondary structure should have low minimum free energy (MFE), indicating stable folding [14]. RNA folding algorithms like Vienna RNAfold used in this study allow for prediction of MFE [15] and, as can be observed in Appendix A, changing the flanking regions to that of miR-708 resulted in an altered MFE with a reduction in MFE for miR-29b and -2467. However, it is generally insufficient to use MFE alone for the detection of stable folding. As a result, more structural features were also analysed, such as seed region accessibility. Inaccessible or highly structured seed regions may reduce miRNA targeting efficiency [16,17]. Design 2 resulted in more linear structures (Figure 1). In addition, accessibility of cleavage sites and the presence of bulges and loops are important for determining processing efficiency [18,19]. DICER enzyme cleaves miRNA precursors at specific sites. These cleavage sites should be accessible within the secondary structure for efficient processing. The more linear structures of design 2 suggested greater accessibility for processing machinery with DROSHA processing sites marked by arrows in Figure 1 that showed a reduced number of bulges and loops especially for miR-29b and -6831 in the areas important for processing machinery. Only MIA PaCa-2 cells were brought forward for further analysis. Following transfection of design 2 into MIA PaCa-2 cells, no levels of mature miRNAs were detectable by RT-qPCR (see figure in Section 2.5). 

### 2.2. An Intrinsic Bias for Processing of the Star form miRNA Could Be Responsible for the Lack of Detectable Expression of miRNA Targets

Mia PaCa-2 cells were examined by RT-qPCR for expression of the passenger strand of miR-2467 (-5p) and miR-708 (-3p) (Figure 2). miR-708 achieved detectable levels of mature miR-708-5p in MIA PaCa-2 cells and was used as a control. miR-708-3p showed no detection and miR-2467-5p had a Ct value of 36. It is well accepted that a Ct value greater than 35 qualifies as non-expressed. The results confirmed that the lack of detection of the mature form of miR-2467-3p was not due to preferential selection for its corresponding passenger strand. 

### 2.3. Examination of Pri- vs. Pre-miRNA Levels in Nuclear vs. Cytoplasmic Fractions of MIA PaCa-2 Cells Showed Precursor Accumulation

The assessment of primary (pri-miRNA) and precursor (pre-miRNA) miRNA levels was conducted using the probes detailed in the methods section, targeting nuclear and cytoplasmic fractions of both parental and overexpression vector-transfected cells. The outcomes validated the successful transfection and transcription of miRNAs, as evidenced by higher levels of pri-miRNAs in the nuclear fractions of transfected cells compared to parental controls, observed through a reduced Ct value for the transfected cells, as shown in Figure 3, Panel A. Endogenous expression levels of parental miRNAs are provided in Appendix A. Intriguingly, despite the majority of these miRNAs not being detectable in MIA PaCa-2 cells, discernible levels of pri-miRNA persisted, indicating that pri-miRNA expression did not display a direct correlation with mature expression. 

Within the cytoplasmic fraction of both parental and transfected cells, pre-miR-708 levels remained undetectable. Notably, pre-miR-708 was the sole miRNA that attained successful maturation. The absence of pre-miR-708 aligned with the hypothesis that this specific miRNA encountered no processing hindrances, thereby leading to the notable miRNA maturation. Conversely, the other miRNAs with no detectable mature miRNA levels demonstrated substantial levels of expression in the precursor form within the cytoplasmic fractions: pre-miR-29b (Ct = 24.51), pre-miR-1290 (Ct = 25.06), pre-miR-2467 (Ct = 29.66), and pre-miR-6831 (Ct = 23.32). This observation suggested the effective processing and export of pri-miRNAs from the nucleus but a subsequent stall in pre-miRNA processing. Our findings demonstrated cytoplasmic accumulation of precursor miRNAs for the subset of miRNAs. By contrast, miR-708, capable of generating detectable mature miRNA, evaded precursor accumulation due to its proficient processing.

These results collectively emphasised the complex interplay between nuclear processing, cytoplasmic transport, and miRNA maturation, shedding light on the distinct fates of individual miRNAs within the cell.

### 2.4. Post DICER-Processed Pre-miRNA Mimics Show Efficient Overexpression

To evaluate the absence of mature miRNAs, we employed pre-miR miRNA mimics, bypassing DICER processing. Our results (Figure 3, Panel B) highlighted the remarkable efficiency of these mimics in achieving a robust abundance of the targeted miRNAs, a feat unattainable through our stable vector approach. Each miRNA mimic led to substantial mature miRNA levels, reaching an impressive RQ of up to 105943 for miR-2467, which exhibited no expression of mature miRNA in parental cells and remained undetectable using both vector designs (refer to Section 2.5). Notably, miR-1290 and miR-6831 demonstrated RQ values of 519.5 and 1535.7, respectively. In comparison, miR-29b, which displayed moderate expression in parental MIA PaCa-2 cells, exhibited the lowest level of overexpression, with an RQ of 86.8. These outcomes underscored the significance of precursor-level dysregulation in hindering miRNA maturation and highlighted the potency of miRNA mimics in addressing this limitation.

**Figure 3 ijms-25-05666-f003:**
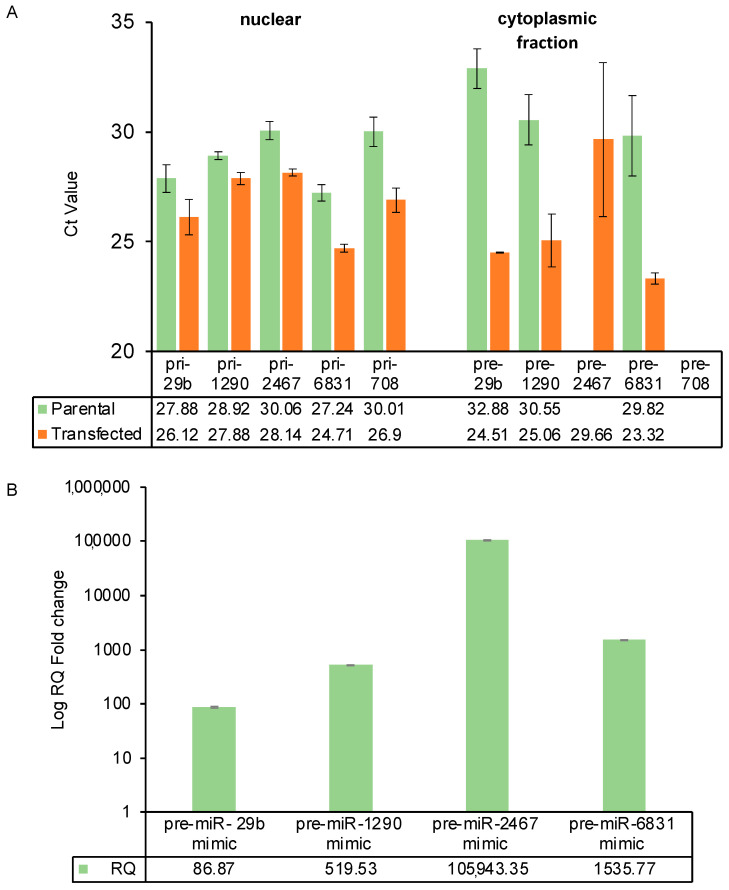
Subcellular localisation and overexpression using pre-miRNAs in MIA PaCa-2 Cells. (Panel **A**): Examining pri- and pre-miRNA expression in nuclear and cytoplasmic fractions reveals the accumulation of precursors specifically in the cytoplasm of MIA PaCa-2 cells. Pri-miRNA probes were used in the nuclear fraction of cells while pre-miRNA probes were used in the cytoplasmic fraction. Data presented as mean and standard deviation of three biological replicates. (Panel **B**): Post-DICER processed pre-miRNA mimics exhibit efficient overexpression, presented as relative quantification (RQ) along with the standard deviation from biological triplicates. Non-transfected MIA PaCa-2 cells served as the calibrator for RQ calculations. Delta Ct calculations used the average of two endogenous controls (miR-16-5p and miR-24-3p). Additionally, miRNA targets without expression in parental cell lines were set with Ct = 40, considering Ct values > 35 as non-expressed.

### 2.5. Investigation of Vector Efficiency in a Panel of Non-Pancreatic Cancer Cell Lines Show Levels of Expression for Some miRNA Targets

To gain a deeper understanding of cell line-specific miRNA processing dynamics, we investigated the relative expression of both vector designs for each miRNA target in a diverse range of cell lines, encompassing non-cancer (HEK293T), non-pancreatic cancer (SK-OV-3), and non-human (CHO-K1) cell lines. Significant fold changes were established at values greater than 2, following established standards [20], taking into account the high expression levels achieved by the control miR-708.

The summarised results for each cell line, including MIA PaCa-2 and PANC-1 (design 1 only), are presented in Figure 4. Notably, within MIA PaCa-2 and PANC-1 cells, only miR-708 exhibited the capability to attain detectable levels of mature miRNA. Design 1 for miR-1290 displayed a modest RQ of 4 in PANC-1 cells, primarily attributed to alterations in endogenous RT-qPCR controls (see methods). Remarkably, design 1 of miR-29b showed mature miR-29b levels in HEK293T (RQ = 2893.8), SK-OV-3 (RQ = 19.4), and CHO-K1 (RQ = 471.4) cells. Furthermore, design 2 of miR-29b facilitated detectable levels in HEK293T (RQ = 3942) and CHO-K1 (RQ = 9) cells.

The control vector, miR-708, achieved discernible expression across all cell lines, with HEK293T exhibiting the lowest level (RQ = 5). Similarly, both designs of miR-1290 achieved miRNA maturation solely in HEK293T cells (design 1 RQ = 7.5, design 2 RQ = 210). It is noteworthy that miR-708 and -1290 displayed the lowest levels of overexpression in HEK293T cells, and their parental endogenous levels were expressed at moderate levels with Ct values of 30.2 and 29.7, respectively (Appendix A). An intriguing observation concerning miR-1290 in HEK293T cells is that while design 1 exhibited lower levels of overexpression, design 2 showed remarkably high levels despite containing the flanking regions of miR-708, which, as mentioned earlier, achieved an RQ value of only 5 in HEK293T cells. In HEK293T cells, miR-29b demonstrated no detectable endogenous levels but achieved high levels of overexpression, with RQ values of 2893.8 and 3942.7 for designs 1 and 2, respectively (Figure 4). For SK-OV-3 cells, although expressed at very low levels just meeting our cut-off for expression with a Ct value of 35 (Appendix A), miR-29b was the sole miRNA to achieve overexpression in this cell line and exhibited the only endogenous expression within the panel. Only design 1 achieved detectable mature miRNA levels, despite miR-708 achieving high levels with an RQ of 913.85 (Figure 4). CHO-K1 was the only cell line that exhibited no endogenous expression (Appendix A) for any of these miRNAs. However, both designs of miR-29b and miR-708 achieved varying levels of overexpression (Figure 4). It is important to note that copy number biasing is an unlikely explanation for the observed differences in RQ values among the cell lines, as transfection efficiency was monitored microscopically by GFP expression and consistently exceeded 75%.

While intriguing trends were evident, the endogenous miRNA expression profiles of the parental cells did not appear to directly correlate with the varying RQ values. Additionally, pri-miRNA structure alone could not be the primary determinant of miRNA maturation, as evidenced by the low RQ value of miR-708 in HEK293T cells and the high RQ value for miR-1290 design 2, which contained the flanking regions of miR-708. Overall, these findings highlighted distinct miRNA expression profiles across diverse cellular contexts, underscoring the intricate interplay between vector design, cell line characteristics, and miRNA target regulation.

## 3. Discussion 

During experiments to overexpress specific miRNAs in human pancreatic cancer cells, our vector designs successfully increased miR-708 expression but not that of miR-29b, -1290, -2467, and -6831. Using Vienna RNAfold, we predicted minimum free energy secondary structures for each pri-miRNA (Figure 1) [15,21]. Bifurcation loops and motifs like mGHG and CNNC are pivotal in miRNA processing, influencing both the efficiency and specificity of biogenesis [22]. Bifurcation loops can impact the accessibility of processing machinery, potentially affecting DROSHA and DICER processing efficiency. The mGHG motif, located in the lower stem region of pri-miRNAs, interacts with DROSHA’s dsRNA-binding domain, facilitating recognition and cleavage of pri-miRNAs [23]. Similarly, the CNNC motif is implicated in miRNA processing regulation [24]. Understanding these motifs is vital for elucidating the complex regulatory mechanisms controlling miRNA processing. Comparing miR-708’s structure with design 1 with that of the other miRNAs, we observed more bifurcation loops (greater than three interior edges with unclear destabilising effects) [25], especially in miR-29b and -1290. Pri-miRNA secondary structures influence processing efficiency, e.g., the mGHG motif in the lower stem interacts with DROSHA’s dsRNA-binding domain [26], which only miR-1290 possessed (Figure 1). This study found that secondary structure alone does not determine processing efficiency, as modification with miR-708’s flanking regions did not overcome the issue.

Comparison of the miR-708 motifs to those of the other four miRNAs (Appendix A) revealed miR-708’s possession of the CNNC motif, which is essential for SRSF3-mediated microprocessor recognition [18]. Previous literature describes this motif at 16–18 nucleotides (nt) from the DROSHA cut site [27]. While miR-708 deviates with a position at 23 nt, miR-29b achieves the optimal positioning, indicating that this motif alone cannot compensate for other processing factors. While miR-708 displays a basal UG motif [28], it sits at -15 and -14 instead of the optimal -14 and -13 [29]. miR-1290 possesses the optimal UG motif placement. miR-29b-1 and -6831-5p exhibited individual U/G nucleotides at the basal junction. DICER cleaves pre-miRNA near the apical loop in the cytoplasm. A UGU motif at the apical junction strengthens DGCR8 interaction, preventing DROSHA displacement and unproductive cleavage [30]. None of the miRNAs, including the successfully overexpressed miR-708, contained a perfect UGU/GUG motif. All miRNAs, except miR-29b, exhibited apical loops shorter than the optimal ≥10 nt [27], with miR-708’s apical loop being the shortest at 7 nt. The data presented here could help inform future chimeric miRNA design principles. 

Numerous factors can influence miRNA expression levels. Methylation changes in regulatory regions, including miRNA genes, are significant in this regard, often associated with dysregulated mature miRNA expression [31]. RNA-binding proteins (RBPs) are pivotal in miRNA biogenesis, with conserved RNA binding domains that recognize specific RNA sequence elements or secondary structures. RBPs also interact with miRISC to regulate miRNA function, modulating miRNA abundance and activity [32]. Post-transcriptional mechanisms like RNA editing and alternative splicing can impact miRNA expression by affecting mature miRNA stability or target specificity. Additionally, competitive endogenous RNAs and RNA decay pathways contribute to miRNA-mediated gene regulation by competing for miRNA binding or facilitating miRNA degradation [33]. Another such regulatory aspect is strand selection, influenced by impaired processing, epitranscriptome modifications (like uridylate addition [34], adenosine-to-inosine deamination [35,36]), and Argonaute strand preference alteration. Typically, the Ago protein takes up either the -5p or -3p miRNA strand, guided by miRNA-targeted mRNA complementarity [37], with one strand predominating as the functional "guide" and the other being degraded [38]. Some miRNAs yield functional mature forms from both strands. To rule out altered strand selection causing undetectable overexpression, we assessed miR-2467’s passenger strand expression alongside miR-708. The results (Figure 2) revealed no detectable expression, ruling out strand selection as the cause for non-detectable mature miRNA levels in these vectors.

We explored three hypotheses to address the absence of mature miRNA levels: (1) Failed DROSHA processing of pri-miRNA? (2) Inadequate transport of pre-miRNA from the nucleus? and (3) Incomplete pre-miRNA processing in the cytoplasm? Our results (Figure 3) identified pre-miRNA accumulation of four miRNAs in the cytoplasm of transfected cells. miR-708, which could produce mature expression, did not exhibit this accumulation, ruling out DROSHA/Exportin 5 dysfunction. While reduced levels of DICER have been described in cancer [39,40], it was unlikely that a global reduction was responsible, as miR-708 produced mature miRNA in MIA PaCa-2. Studies have shown that DICER protein levels do not always correspond to mature miRNA levels [41], potentially due to non-canonical processing pathways that can operate independently of components like DGCR8, DICER, Exportin-5, or Argonaute 2. With cytoplasmic precursor miRNA accumulation established in MIA PaCa-2 cells, we explored these miRNAs in additional cell lines: SK-OV-3 (non-pancreatic), HEK293T (non-cancerous), and CHO-K1 (non-human) (Figure 4). The differences in fold change between designs 1 and 2 indicated that while these processing complexities are not solely determined by structure or sequence, they are important. Endogenous miRNA expression in parental cells (Appendix A) and transfection efficiency were also ruled out as principal factors contributing to the different RQ values observed between cell lines (Appendix A). 

Unlike stable vectors, pre-miR mimics resulted in abundant levels of mature miRNAs (Figure 3). Dysregulated RISC assembly can hinder miRNA synthesis [42]. Pre-miR miRNA mimics, chemically modified RNA molecules mirroring endogenous miRNAs, bypass DICER processing, directly engaging RISC. Our demonstration that bypassing DICER processing leads to mature miRNA levels suggested the issue is not within RISC. Some miRNAs are subject to negative feedback regulation. When the levels of a particular mature miRNA increase (e.g., due to disease, external stimulus, or treatment), it can trigger regulatory mechanisms that aim to counteract this increase, such as transcriptional alteration of the corresponding pri-miRNAs. It is interesting that these miRNAs were identified as being upregulated in either normal versus tumour or tumour versus F1 in pancreatic cancer models (Appendix A) but were either lowly expressed or undetected in the pancreatic cancer cell lines. While this intriguing point invites further exploration in future experiments, offering a promising avenue for deeper insights into miRNA regulation, we believe that our results of miRNA maturation when endogenous processing was skipped by using pre-miRNA mimics (Figure 3) underscores a dysfunction post-DROSHA but pre-DICER/RISC and dispels concerns about competing endogenous RNAs (ceRNAs) sequestering our miRNAs [43]. 

This study has demonstrated the distinct behaviours of different miRNAs with the same vector in the same cell line, the same miRNA between the two vector designs, and with the same miRNA across different cell lines (Figure 4). This diversity in expression patterns in an miRNA- and cell line-specific manner, along with its basis in impaired pre-miRNA processing, has not been reported previously. Our findings also show that miRNA mimics are a good alternative, while more studies are required to fully understand the complex processing dynamics that can affect stable vector efficiency. These findings make a significant contribution to the field of miRNA cellular engineering and point to a much greater degree of complexity in the processes controlling miRNA maturation than previously appreciated. 

## 4. Materials and Methods

### 4.1. Cell Culture

The human pancreatic cell lines MIA PaCa-2 (ECACC, Salisbury, UK) and PANC-1 (ATCC, Rockville, MD, USA) were cultured in DMEM high glucose (D5671, Merck, Darmstadt, Germany) supplemented with 5% (*v*/*v*) foetal calf serum and 2% (*v*/*v*) l-glutamine (2503008, Thermo Fisher, Waltham, MA, USA); the human ovarian cancer cell line SK-OV-3 (gifted from the Collins group at NICB, DCU-passage 40) was cultured in RPMI-1640 (Sigma-Aldrich, St. Louis, MO, USA, R8758) supplemented with 10% (*v*/*v*) foetal bovine serum (FBS); the human embryonic kidney cell line HEK293T (gifted from the Walsh lab group at NICB, DCU-passage 39) was cultured in DMEM-high glucose (D5671, Merck, Darmstadt, Germany); and the Chinese Hamster Ovary cell line CHO-K1 (ATCC, CCL-61) was cultured in DMEM-F12 (Sigma-Aldrich, D3487) supplemented with 5% (*v*/*v*) FBS. All cell lines had been STR profiled (IDEXX BioAnalytics, Kornwestheim, Germany). Cells were grown in a humidified atmosphere with 5% CO_2_ at 37 °C in T-25 and T-75 culture flasks. Cell lines were split (1:5) using PBS to wash and trypsin-EDTA (Thermo Fisher, Waltham, MA, USA) when 80–90% confluency was reached. 

### 4.2. miRNA Overexpression Vector Design

For the generation of both vector designs (Figure 5), the pTight-TET-ON vector (pcDNA3.1(+)HYG) expressing a destabilised GFP reporter (d2eGFP) was used as a backbone vector. For design 1, the DNA fragments containing the precursor miRNA (pre-miRNA) and 70 bp genomic flanking regions on both 5′ and 3′ ends (pri-miRNA) were cloned downstream of d2eGFP, resulting in the conventional miRNA expression vector. A detailed description of vector construction has been described previously [44]. Design 1 was constructed for each miRNA among a list of priority miRNAs, including miR-21, -29b, -125a, -148, -206, -378e, -615, -708, -1290, -2467, and -6831. For design 2, the DNA fragments containing the precursor miRNA (pre-miRNA) were cut out using restriction enzymes NotI and EcoRI (Thermo Fisher, Waltham, Massachusetts, USA) and cloned into the genomic flanking regions of miR-708-5p by complementary sticky ends. miR-708 was chosen as the positive control miRNA for all future analysis as it achieved the highest level of overexpression. All sequences are provided in Appendix A. 

### 4.3. Transfection 

Cells were counted and seeded in 6-well plates and allowed to attach for 24 h before transfection. The seeding densities were as follows: MIA PaCa-2 and DLKP were seeded at 4 × 10^4^ cells total, PANC-1 and SK-OV-3 at 5 × 10^4^ cells total, and CHO-K1 at 3 × 10^4^ cells total. For stable vectors, a total of 1 μg DNA was mixed with 250 μL Dulbecco’s Modified Eagle Medium F12 and 2 μL TransIT X2 reagent (MIR6000, Mirus Bio, Madison, WI, USA) and incubated for 30 min at RT. Cells were then placed in fresh media and 250 ul of the TransIT-X2-DNA transfection complexes was added to each well. A negative control without DNA was also included. Two days post transfection, cells were induced with 100 ng/mL doxycycline and transfection efficiency was confirmed 24 h post induction through GFP assessment by fluorescence microscopy inspection. Transfection efficiencies, analysed by GFP expression and doxycycline toxicity assays, are shown in Appendix A. Stable overexpressing cell lines were selected and generated by supplementing with 400 ng/mL Hygromycin B (Roche, Basel, Switzerland). Chemically synthesised linear double-stranded pre-miR mimics (AM17100, Thermo Fisher, Waltham, Massachusetts, USA) were used to test miRNA maturation without DICER processing. For transfection with these mimics, a working concentration of 10 μM was mixed with 3 μL TransIT X2 reagent in 100 μL Dulbecco’s Modified Eagle Medium F12. The mixture was left for 30 min at room temperature and then added to the cells. 

### 4.4. RNA Isolation and cDNA Synthesis

One day prior to RNA collection, cells were induced with 100 ng/mL doxycycline (Merck, Darmstadt, Germany) and GFP expression was confirmed by fluorescence microscopy. RNA samples were collected from 80–90% confluent 6 well plates. Cell pellets were collected in a total of 1 mL TRIzol (15596018, Thermo Fisher, Waltham, MA, USA) and stored at −80 °C until processed. Total RNA was isolated following the manufacturer’s recommendations. RNA quality and quantity were assessed using a NanoDrop (Thermo Scientific, Waltham, Massachusetts, USA). In order to remove contaminating DNA, DNase I treatment (AMPD1-1KT, Merck, Darmstadt, Germany) was performed following the manufacturer’s instructions. 

### 4.5. RT-qPCR 

For mature miRNAs, samples were prepared using the TaqMan MicroRNA Reverse Transcription Kit (ThermoFisher, 4366596), as per the manufacturer’s recommendations. For primary and precursor miRNA detection, samples were prepared using the High Capacity cDNA Reverse Transcription Kit (ThermoFisher, 4368814). The ddCt method was used to determine the relative quantification (RQ) of each miRNA with miR-16-5p and miR-24-3p used as endogenous controls [13]. Where parental levels of an miRNA were undetectable, the Ct value was set to 40 for completion of the RQ calculation as it is well accepted that a Ct greater than 35 is considered non-expressed. The RQ of mature miRNAs was assessed from a total of 10 ng cDNA samples using the TaqMan® Advanced miRNA Assay (ThermoFisher, A25576), as per the manufacturer’s recommendations. The RQs of primary and precursor miRNAs were assessed from a total of 100 ng cDNA sample using the TaqMan^®^ Pri-miRNA Assay (ThermoFisher, 4427012) and TaqMan Non-coding RNA Assay (ThermoFisher, 4426961), as per the manufacturer’s recommendations.

## Figures and Tables

**Figure 1 ijms-25-05666-f001:**
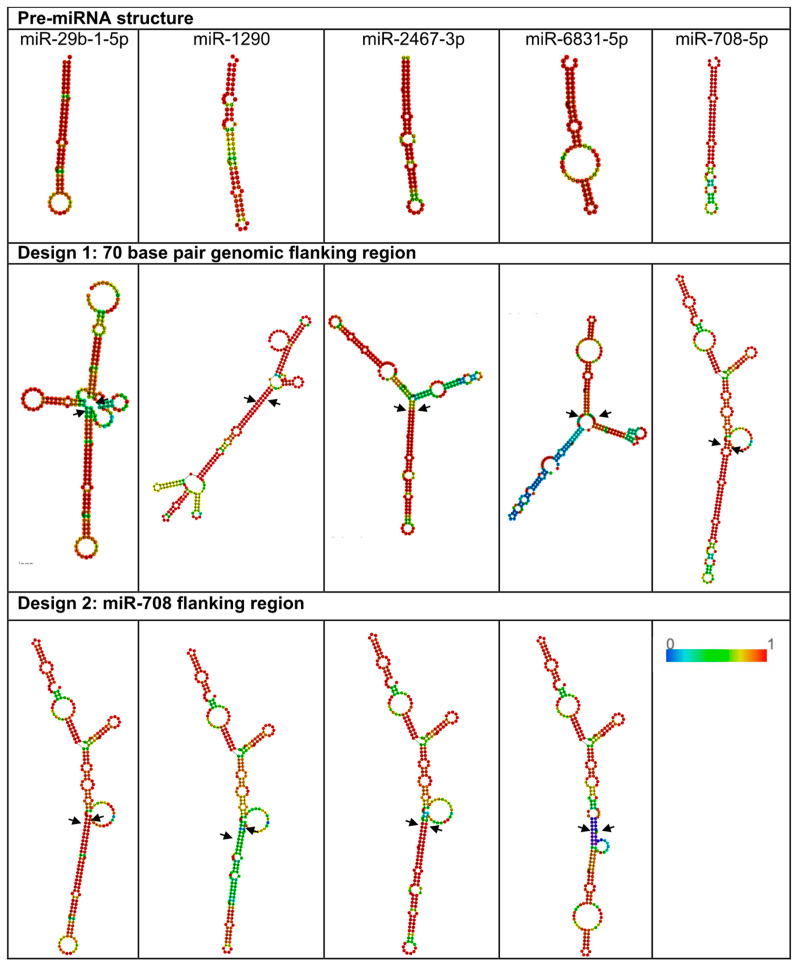
Predicted RNA secondary structures for vector designs 1 and 2. Row 1 shows the pre-miRNA structures for miR-29b, -1290, -2467, -6831, and -708 generated using Vienna RNAfold server. Each column contains the corresponding secondary structure for each miRNA target with endogenous 70 base pair flanking regions (design 1) and miR-708 flanking regions (design 2). Arrows indicate DROSHA processing sites and colours indicate base pairing probability from 0-1 as shown in the scale bar with 1 being most probable base pairing.

**Figure 2 ijms-25-05666-f002:**
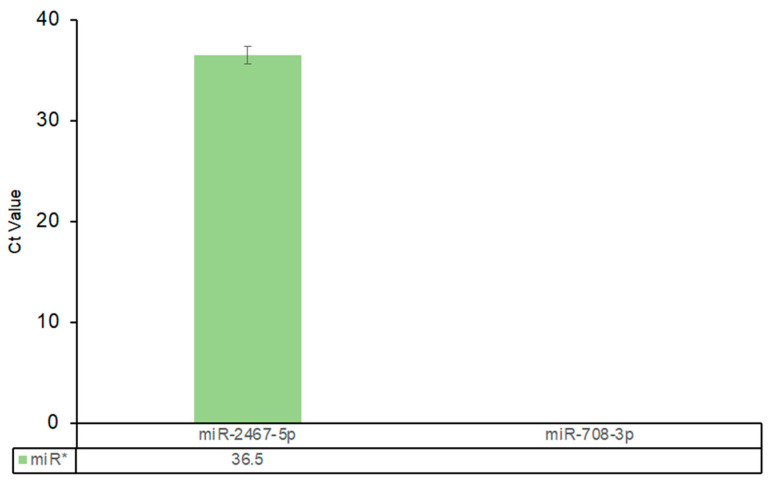
Evaluation of miRNA* expression levels shows preferential processing of the star form is not responsible for the lack of detection in MIA PaCa-2 cells. The data are presented as mean and standard deviation.

**Figure 4 ijms-25-05666-f004:**
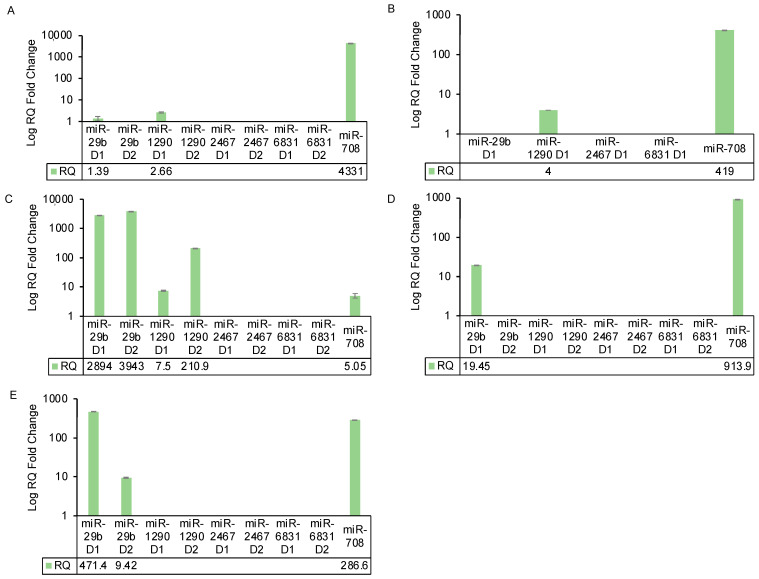
Investigation of vector efficiency in a panel of non-pancreatic cancer cell lines show levels of expression for some miRNA targets. The data are presented as relative quantification (RQ) and standard deviation of biological triplicates. For RQ non-transfected MIA PaCa-2 was used as the calibrator. For Delta Ct calculations, the average of two endogenous controls (miR-16-5p and miR-24-3p) was used, except for SK-OV-3 cells, where 16-5p was used alone due to changes in 24-3p. For RQ calculations, the miRNA target with no expression in parental cell lines was set as Ct = 40, as anything with a Ct value > 35 is considered non-expressed. Panel description: (**A**) MIA PaCa-2, (**B**) PANC-1, (**C**) HEK293T, (**D**) SK-OV-3, (**E**) CHO-K1.

**Figure 5 ijms-25-05666-f005:**
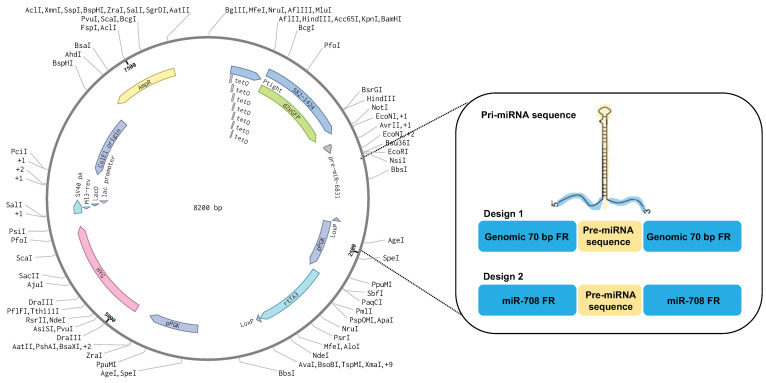
Schematic of two stable vector designs for the overexpression of miRNAs. Design 1 shows a conventional primary miR (priMIR) expression system consisting of 70 bp genomic flanking region (FR) surrounding the miR transcript unit under the pTight promoter, part of an inducible TET-ON system that requires the presence of doxycycline to activate expression, along with co-expression of the pre-miRNA with a fluorescent protein (destabilised GFP) to follow the induction of the system. Design 2 is composed of the same vector backbone, but the miRNAs of interest were placed within the 76 bp flanking region of miR-708, which showed successful overexpression in vitro.

## Data Availability

Data is contained within the article.

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
