# Peer review of "miRNA- and Cell Line-Specific Constraints on Precursor miRNA Processing of Stably Transfected Pancreatic Cancer and Other Mammalian Cells"

_ijms, 2024, doi:10.3390/ijms25115666_

Round 1

Reviewer 1 Report

Comments and Suggestions for Authors

General comments:

These findings make a significant contribution to the field of miRNA cellular engineering and point to a much greater degree of complexity in the processes controlling miRNA maturation than previously appreciated.

Major comments:

1. “cell-specific” only appears one time at the title. I cannot find it in the context. It also unclear for the results or background for “cell-specific”. Please enhance the background and description in abstract, Introduction, results, and discussion.

2. Title: ..stably transfected cells. But the material shows that it is pancreatic cancer cells. It is suggest to change title to “.. stably transfected pancreatic cancer cells.

3. Provide criteria used to assess the predicted secondary structures of the designed miRNAs and how structures may influence miRNA processing.

4. Please provide how a two-fold increase was seen in pri-miRNA levels in the nuclear fractions of transfected cells.

5. Discuss how other factors may influence miRNA expression levels, such as epigenetic modification, RNA-binding proteins and post-transcriptional mechanisms.

6. Discuss the significance of bifurcation loops and motifs such as mGHG and CNNC in miRNA processing

Minor comments:

1.Result 3.1 and 3.2 Titles: Please change the “interrogative sentence” to “Affirmative sentence”.

Author Response

Dear Reviewer,

We would like to express our sincere appreciation for the diligent review and insightful feedback provided on our manuscript titled " Mapping miRNA-and cell-specific constraints on precursor miRNA processing of stably transfected cells." We are grateful for the opportunity to address your comments and enhance the quality of our work.

Below, we outline our responses to each of your comments:

**Reviewer 1:**

**Major Comments:**

  1. **Title Enhancement:** We appreciate the reviewer's suggestion to clarify the focus of the paper in the title. We have amended the title to better reflect the cell specificity addressed in the study. Additionally, we have revised the abstract, results, and discussion sections to provide a more comprehensive overview of the study, emphasising the observed cell line specificity in miRNA processing efficiency (Lines 51-55, 63-65, 312, 468-470).

  1. **Criteria for Assessing Secondary Structures:** We have expanded the descriptions of our criteria for assessing secondary structure (Lines 170-187), accompanied by an additional table (Table IV) in the supplementary data.

  1. **Explanation of Two-Fold Increase in pri-miRNA Levels:** The explanation for the differences in levels of expression for the pri-miRNA has been reworded to provide more clarity for the reader (Lines 224-227).

  1. **Discussion of Other Factors Influencing miRNA Expression Levels and the Significance of Bifurcation Loops and Motifs:** We have expanded on the discussion regarding these important points (Lines 373-380 and 405-415).

**Minor Comments:**

  1. **Result Titles:** These changes have been made as requested (Lines 155-156, 201-202).

Thank you for your time and consideration.

Sincerely,

Taylor J Allen Coyle

Reviewer 2 Report

Comments and Suggestions for Authors

The authors Taylor J Allen-Coyle et al delves into altering of miRNA expression in pancreatic cancer cells and the findings underscore the potential of miRNA mimics in overcoming complexities in therapeutic interventions.

Minor comments:

1) P2, line 80 - The author used miR-708 as positive control, it would be great if author provide specific reason for using it as positive control.

2) P3, line 91 to 94 - SK-OV-3 and HEK293T cells lines were gifted from other labs, the author needs to provide about passage no and authentication methods for reproducing the results.

3) P4, line 130 - The author needs to provide about transfection efficiency and any potential cytotoxic effects 

4) References - The authors need to provide latest references.

Author Response

Dear Reviewer,

We would like to express our sincere appreciation for the diligent review and insightful feedback provided on our manuscript titled " Mapping miRNA-and cell-specific constraints on precursor miRNA processing of stably transfected cells." We are grateful for the opportunity to address your comments and enhance the quality of our work.

Below, we outline our responses to each of your comments:

**Reviewer 2:**

**Minor Comments:**

  1. **Explanation for Using miR-708 as Positive Control:** We have clarified the specific reasoning for using miR-708 as a positive control (Lines 145-149).

  1. **Details on Cell Lines:** The methods section has been edited to include the passage number of these cell lines and to clarify that all cell lines used in this study underwent STR profiling methods section 4.1.

  1. **Transfection Efficiency and Cytotoxic Effects:** This has now been addressed with the addition of a Figure in the supplementary data showing GFP microscopy and doxycycline toxicity assays and highlighted in methods section 4.3.

  1. **Update References:** New references have been added relevant to the added discussion points.

We believe these revisions significantly strengthen the manuscript and address the reviewers' concerns comprehensively. We hope you find these changes satisfactory for consideration for publication in in the special issue “Targeted Delivery of Nucleic Acids” in International Journal of Molecular Sciences.

Thank you for your time and consideration.

Sincerely,

Taylor J Allen Coyle